# Safety and Efficacy of Bevacizumab in Cancer Patients with Inflammatory Bowel Disease

**DOI:** 10.3390/cancers14122914

**Published:** 2022-06-13

**Authors:** Ruth Gabriela Herrera-Gómez, Miruna Grecea, Claire Gallois, Valérie Boige, Patricia Pautier, Barbara Pistilli, David Planchard, David Malka, Michel Ducreux, Olivier Mir

**Affiliations:** 1Department of Cancer Medicine, Gustave Roussy Cancer Institute, 94805 Villejuif, France; alinamiruna.grecea@gustaveroussy.fr (M.G.); valerie.boige@gustaveroussy.fr (V.B.); patricia.pautier@gustaveroussy.fr (P.P.); barbara.pistilli@gustaveroussy.fr (B.P.); david.planchard@gustaveroussy.fr (D.P.); david.malka@gustaveroussy.fr (D.M.); michel.ducreux@gustaveroussy.fr (M.D.); mir.olivier@gmail.com (O.M.); 2Oncology Department, Lausanne University Hospital, University of Lausanne, 1011 Lausanne, Switzerland; 3Oncology Institute ‘Prof. Dr. I. Chiricuta’, 400015 Cluj-Napoca, Romania; 4Department of Gastroenterology and Digestive Oncology, Hôpital Européen Georges Pompidou, Assistance Publique–Hôpitaux de Paris, 75015 Paris, France; claire.gallois2412@gmail.com

**Keywords:** inflammatory bowel disease, ulcerative colitis, Crohn’s disease, bevacizumab, chemotherapy, cancer

## Abstract

**Simple Summary:**

In patients with inflammatory bowel disease, chronic inflammation is a risk factor for the development of digestive and nondigestive cancers. The treatment, as in patients without inflammatory bowel disease, is a combination of chemotherapy and targeted treatments, such as bevacizumab and more recently, immunotherapy. It is generally believed that the use of bevacizumab and chemotherapy could increase toxicity and lead to adverse events in this population. This study aims to evaluate the safety and efficacy of the combination of bevacizumab and chemotherapy in patients with quiescent or moderately active inflammatory bowel disease in various forms of cancer and by increasing the quality of patient care in this subgroup.

**Abstract:**

Background: The safety of bevacizumab in combination with chemotherapy in patients with inflammatory bowel disease (IBD) and digestive and nondigestive cancers is poorly documented. Methods: We retrospectively evaluated patient records of all adult cancer patients with IBD at our institution from April 2007 to May 2016 with an update in November 2019. Results: Twenty-seven patients with a history of IBD (Crohn’s disease, *n* = 22; ulcerative colitis, *n* = 5) who were treated with bevacizumab and chemotherapy for metastatic solid tumors were identified. At the time of advanced cancer diagnosis, 18 patients had quiescent IBD, whereas 9 patients had moderately active IBD. Among those with moderately active IBD, five had received corticosteroids less than six months prior to cancer diagnosis and one had received infliximab. The treated cancers were colorectal cancer (*n* = 13), small bowel cancer (*n* = 4), non-small cell lung cancer (*n* = 3), breast cancer (*n* = 3), and other cancers (*n* = 4). Patients received bevacizumab in combination with chemotherapy and/or as maintenance for a median of 6.7 months. Grade 2 or higher bevacizumab-related complications were proteinuria in two patients and hypertension, diarrhea, rectal bleeding, and intestinal perforation in one patient each. No clinical IBD flares were observed during bevacizumab treatment. Conclusion: Bevacizumab combined with chemotherapy is safe in cancer patients with moderately active or quiescent IBD.

## 1. Introduction

*Preliminary results of this investigation have been presented at the annual ASCO GI meeting 2019 [1]. After thoughtful review of data in November 2019, one patient of the initially published cohort was removed because he did not meet the inclusion criteria*.

Ulcerative colitis (UC) and Crohn’s disease (CD) are two conditions commonly referred to as idiopathic inflammatory bowel disease (IBD). The diagnosis of IBD is typically established in young adults and is based on histological findings. In the case of UC, the digestive mucosal tissue of the large bowel is affected. Usually, lesions are arising from the rectum having a continuous extension. In CD, a transmural inflammation is observed that may affect any part of the gastrointestinal tract but typically develops in the terminal ileum.

Research suggests that the pathogenesis of IBD results from an inappropriate inflammatory response to intestinal microbes in a genetically susceptible host [2]. There are different reports stressing the critical role of neo-angiogenesis in IBD eventually driven by inflammation and the immune response, a so-called “immune angiogenesis” [3]. In patients with IBD, a higher density of micro vessels in the intestinal tissue has been demonstrated, as well as higher levels of vascular endothelial growth factor (VEGF) in serum and colonic tissue [4].

In Europe and North America, the prevalence of IBD ranges from 0.3% to 0.5% in the general population. A stable or decreasing incidence over the past decades is observed. An increase in both prevalence and incidence of IBD has been observed in newly industrialized countries [5].

Manifestations of IBD range from mild, asymptomatic inflammation to extensive inflammation of the gastrointestinal (GI) tract. IBD presents frequently with extra intestinal manifestations that may occur during active IBD, such as erythema nodosum and peripheral arthritis [6]. Other complications, such as ankylosing spondylitis, iritis, and primary sclerosing cholangitis, may develop independently of IBD activity.

Consensual definitions of disease severity for IBD are lacking. Research suggests grading the disease by the impact of the disease on the patient, the inflammatory burden, and the disease course, thus permitting a classification of quiescent, moderately active, and severe disease [7].

Therapies for IBD may consist of untargeted therapies, such as aminosalicylates, corticosteroids, immunomodulators, and targeted therapies, including biologics combined with other general measures, such as diet programs. Additionally, surgical procedures are proposed if necessary. In non-responders further but still experimental therapies are emerging, involving targeted therapies with small molecules, apheresis therapy, interventions that aim to improve the intestinal microbiota, cell therapy, and exosome therapy [8].

Because of chronic inflammation, patients diagnosed with IBD have a slightly increased risk of developing cancer compared to the general population. Of note, the lifetime risk of developing colorectal cancer (CRC) in patients with IBD in Europe and North America is 1.5 to 2 times higher. Most important risk factors for CRC in patients with IBD are extensive colitis and chronic active disease [9].

Immunosuppressive therapies and lifestyle factors, such as tobacco consumption, frequently observed in patients with CD may further contribute to an elevated risk of cancer [9]. In addition to CRC, other digestive cancers, such as adenocarcinoma of the small intestine, cholangiocarcinoma, intestinal lymphomas, and anal cancers, may be related to IBD. Cancers associated with immunosuppression are not only hematologic cancers but also skin cancers, such as melanoma cancer, basal cell carcinomas, and squamous cell carcinomas, as well as urinary tract cancers [9]. It goes without saying that independently of possible IBD-related cancers, sporadic cancers remain the most frequent cancers in patients with IBD.

Although no recommendations on treatment approaches for IBD-associated cancer have been published, the same treatments used for sporadic cancers may be ideal for IBD-associated cancer. These treatment options include surgery, chemotherapy, radiotherapy, immunotherapy, and targeted treatments, such as VEGF inhibitors.

Concerns regarding the administration of chemotherapy in patients with IBD are based on a suspected increase in gastrointestinal side effects, such as diarrhea, as observed in a small retrospective study [10]. Thus, special caution is needed when chemotherapy regimens known for their enteric side effects, such as 5-fluorouracil (5-FU) and irinotecan, are used [10]. Interestingly, chemotherapy alone, as opposed to chemotherapy in combination with hormonal therapy or hormonal therapy alone, showed lower rates of IBD reactivation in patients with various solid malignancies, such as breast, lung, GI, genitourinary, and soft tissue cancer in a recent retrospective study [11].

UC exacerbation has been described not only in patients receiving sunitinib and sorafenib, two oral multiple kinase inhibitors targeting not only the VEGF pathway, but also other receptors as well, such as platelet-derived growth factor receptor alpha (PDGFRA), which are potentially involved in the pathophysiology of IBD [12,13]. An increased risk of GI side effects also has been reported in patients treated with immune checkpoint inhibitors [14].

Bevacizumab is a humanized IgG1 monoclonal antibody directed against VEGF-A and has been proven to be effective in various cancers, such as metastatic CRC, recurrent or metastatic non-small cell lung, ovarian cancer, advanced breast cancer, glioblastoma, and cervical cancer. Bevacizumab has been approved by the FDA in 2004 and the EMA in 2005 for the treatment of advanced CRC. Side effects of bevacizumab may include reduced wound healing and bleeding, as well as intestinal perforation and surgical anastomosis leak [15,16].

Owing to the sparse literature and studies in this context, safety concerns regarding bevacizumab are currently subject to scientific debate. The aim of our study is to evaluate the safety profile of bevacizumab in cancer patients with a history of IBD.

## 2. Patients and Methods

### 2.1. Patients

We retrieved the electronic records of all patients with known IBD treated for GI and extra-GI cancers at Gustave Roussy Cancer Institute between April 2007 and May 2016. Gustave Roussy is a tertiary cancer center that acts as a referral center for patients in the greater Paris metropolitan area, treating over 12,000 newly diagnosed cancer patients each year. Each patient admitted to Gustave Roussy is systematically assessed for comorbidities to prevent potential complications of systemic treatments. Pre-existing conditions, such as IBD, are recorded. For this study, two researchers (GH/MG) independently evaluated the records of all patients.

### 2.2. Variables

Assessment of IBD activity at cancer diagnosis was performed by an experienced gastroenterologist according to individual disease burden, inflammatory parameters, and variables of disease course, including a number of disease flares and disease extension. We defined moderately active IBD by compatible endoscopic or histologic findings up to six months before cancer diagnosis or as symptomatic disease characterized by diarrhea, weight loss, abdominal pain, blood or mucus in stool, or extraintestinal manifestations. Quiescent IBD was defined as disease remission observed on endoscopy and confirmed histologically up to six months before cancer diagnosis or asymptomatic disease [11]. The patients’ characteristics were evaluated, including age, sex, performance status (PS), and acquired risk factors, such as tobacco use. Moreover, the characteristics of each patient’s IBD were documented, including the type of IBD, age at diagnosis, type of intestinal manifestation of IBD, and previous surgical treatments for IBD. Histopathological diagnosis, mutational status in patients with CRC, age at cancer diagnosis, and cancer treatments, including radiation or surgery, were also assessed.

We defined a positive smoking history in patients meeting the definition of current or former smoker according to the guidelines of the Centers for Disease Control and Prevention (CDC). Patients under 18 years of age at the time of cancer diagnosis, as well as those treated with chemotherapy alone were excluded from the study. Moreover, patients with central lung cancers, active bleeding, brain metastases, anticoagulant treatment, and poor performance status (>3) were considered ineligible for this treatment because of the risk of bleeding induced by the use of bevacizumab; thus, they were also excluded from the study [17]. Other exclusion criteria were as follows: grade ≥1 proteinuria, uncontrolled hypertension before initiation of treatment, renal dysfunction, wound healing disorders, and recent arterial thromboembolic events (within 6 months) that contraindicated the use of bevacizumab. Finally, patients enrolled in any clinical trial were excluded.

### 2.3. Data Sources

Radiological examination was performed by computed tomography (and/or magnetic resonance imaging) every 8–12 weeks or more often if clinically indicated and evaluated according to RECIST v1.1 [18]. Toxicities and administration of treatments were recorded at each visit and were rated according to CTCAE v4.03.

### 2.4. Ethical Guidelines

This study was approved by the Gustave Roussy Institutional Review Board and was conducted in accordance with ethical clinical practices and applicable laws. The requirements of abiding by the Declaration of Helsinki and receiving informed consent were waived due to the retrospective nature of the study.

### 2.5. Statistical Methods

Descriptive statistics (median, range, and 95% confidence intervals (CI)) were used to analyze patients, characteristics of IBD, and treatment-induced toxicities. The follow-up period for all patients started at the time of cancer treatment initiation. Patients were censored at the date of disease progression, last contact, or death. Progression-free survival (PFS) was calculated from the date of first administration of treatment until disease progression or death from any cause. Survival analysis was performed using the Kaplan-Meier method based on disease activity. Toxicity and activity data were collected on October 31, 2019. The calculations were performed using NCSS 2020 software (Kaysville, UT, USA).

## 3. Results

### 3.1. Characteristics of the Study Population

We identified 68 patients with IBD who were treated for cancer at our institution. Among them, 27 patients (CD: 22 patients [81%]; UC: 5 patients [19%]) with GI or extra-GI cancer treated with bevacizumab and cytotoxic chemotherapy were identified (Table 1). Forty-one of the sixty-eight (60%) patients were excluded as they were treated with chemotherapy alone. Sixteen patients included in the study (59%) were women. The mean age at IBD diagnosis was 30.8 years (range: 9–57).

The most frequently documented intestinal manifestation was pancolitis in two patients with UC (40%) and in five patients with CD (23%). Tobacco consumption was documented in five patients (19%). Eighteen (67%) patients presented with quiescent IBD, and nine patients (33%) presented with moderately active IBD at the time of cancer diagnosis. The documented IBD treatment in the 6 months before cancer diagnosis was systemic corticosteroids in five patients and infliximab in one patient. 5-ASA and topical corticosteroids were additionally used as clinically indicated. Twenty-one patients were off systemic treatment and notably no high-dose corticosteroids (defined as prednisone equivalent of >20 mg/d) were initiated during cancer therapy.

The median age at cancer diagnosis was 45 years (range: 23–73). The mean time between IBD diagnosis and cancer diagnosis was 15 years (range: 0.5–39) among the whole study population, 18 years (range: 6–39) among patients with quiescent IBD at cancer diagnosis, 11 years (range: 0.5–19) among patients with moderately active IBD at cancer diagnosis (Table 2), 13 years (range: 0.5–23) among patients with CD, and 24 years (range: 12–39) among patients with UC (Table 3).

According to the Zubrod scale, 92% of patients had a performance status of 0 or 1. The most common primary tumor site was CRC in 13 patients (48%), comprising eleven patients with CD and two patients with UC (Table 3). Other frequent cancers were breast cancer and lung cancer in three patients each (11%). The remaining cancers in the study population were thyroid cancer, glioblastoma, and cancer of unknown primary (CUP) in one patient each (3.5%). Two of the 13 patients with CRC had *KRAS* mutations.

Surgery of the primary tumor was performed per protocol according to the disease. Finally, bevacizumab treatment was administered until disease progression, unacceptable toxicity, or the decision to suspend treatment by the patient or treating physician.

### 3.2. Results of the Whole Study Population

Eighteen patients (67%) had quiescent IBD at cancer diagnosis and 22 patients (81%) had CD.

Concomitant cancer treatment consisted of 5-FU (n = 18; 67%) in combination with oxaliplatin (n = 8), irinotecan (n = 8), or both (n = 1); taxanes (n = 7; 26%); lomustine (n = 1); and everolimus (n = 1). Combination therapy with bevacizumab was 70% with a median treatment time of 6.7 months for the first line of treatment and 30% for the second line of treatment (Table 2).

The reasons for treatment discontinuation were progression in 21 patients (78%) and toxicity in three patients (11%). Among the remaining three patients, follow-up was insufficiently documented in one patient, one patient continued with surgery and intraperitoneal chemotherapy after reaching a favorable response, and finally, one patient with CRC died because of intestinal perforation. The most frequent bevacizumab-related toxicities were hematological toxicities in three patients (11%), proteinuria in two patients (7.4%), and epistaxis, oral fistula, rectal bleeding, diarrhea, intestinal perforation, and arterial hypertension in one patient each (Table 4). No IBD flares were observed and no new IBD treatment was required during bevacizumab treatment.

### 3.3. Adverse Events in Patients with Moderately Active Inflammatory Bowel Disease

In the subset of patients with moderately active IBD who were treated with bevacizumab, grade 2 proteinuria occurred in one patient, which resolved spontaneously upon discontinuation of treatment. Grade 2 epistaxis and grade 2 oral-gingival fistula occurred in another patient. These symptoms also resolved upon treatment suspension. Additionally, one patient presented with grade 2 diarrhea requiring medical treatment, and the cause of this complication was presumed to be irinotecan treatment. Once the symptoms resolved, the patient continued treatment with 5-FU and bevacizumab without symptom recurrence (Table 4).

### 3.4. Adverse Events in Patients with Quiescent Inflammatory Bowel Disease

Among patients with quiescent IBD, one patient presented with grade 2 hypertension. In this case, treatment was discontinued independently due to disease progression. Furthermore, grade 3 proteinuria occurred in one patient necessitating treatment suspension. After resolution of proteinuria, the treatment could be resumed. A third patient experienced grade 2 rectal bleeding that was investigated using colonoscopy. After exclusion of disease reactivation, treatment was continued. Finally, intestinal perforation secondary to mesenteric ischemia resulted in the death of one patient (*discussed below*).

### 3.5. Digestive Perforation during Treatment with Bevacizumab and Chemotherapy

We observed grade 5 intestinal perforation in a 43-year-old female patient with metastatic colon cancer diagnosed 9 years earlier with frequently relapsing pancolic CD. The patient’s frequently relapsing IBD required previous treatment consisting of corticosteroids, as well as infliximab up to six months before diagnosis. Subtotal colectomy was performed followed by first-line administration of FOLFOX chemotherapy without bevacizumab for a total of 4 months. Later, bevacizumab was administered as maintenance therapy. Four weeks after the initiation of maintenance treatment, the patient presented with an acute abdomen with septic shock. Based on the computed tomography results, intestinal perforation, possibly because of mesenteric ischemia, was suspected. Unfortunately, the patient did not respond to treatments, was ineligible to surgery, and deceased rapidly. An autopsy was not performed. We believe this complication may be related to bevacizumab treatment.

### 3.6. Results of Patients Treated for Colorectal Cancer

Ten patients with metastatic CRC were treated with bevacizumab in combination with chemotherapy in the first-line setting, and another three received this treatment in the second-line or later. The median PFS was 5.4 months (95% CI: 4.0–7.6) in patients in the first-line setting (Figure 1). Bevacizumab was well-tolerated in ten patients (85%) with CRC, while complications were observed in three patients: two patients presented with grade 2 rectal bleeding and one patient deceased because of intestinal perforation (*see above*).

## 4. Discussion

Patients with IBD are at an increased risk of developing cancer. Due to the fact that potential side effects of chemotherapy in combination with anti-angiogenic treatments, such as bevacizumab, are poorly investigated, treatment may be especially challenging in these patients.

Moreover, the concern about the potential side effects of bevacizumab, including intestinal perforation, in patients with IBD and GI cancer is based on observations that bevacizumab may cause tumor necrosis in the GI wall. More specifically, this might lead to bowel perforation [15]. Furthermore, bevacizumab might affect the GI vascularization, favoring ischemic perforation in the normal bowel and in digestive anastomosis [19]. Bevacizumab treatment could also prevent the healing of GI ulcers or colonic diverticulitis by reducing wound healing. In addition, it contributes to the formation of peptic ulcers, which are worsened by treatment with steroids commonly used for IBD [20].

Contrary to these expected reactions, the results of our retrospective study investigating the activity and potential side effects of bevacizumab in combination with chemotherapy in patients with quiescent or moderately active IBD demonstrated good clinical tolerability.

In our cohort, comprising patients with intestinal and extraintestinal cancers, toxicities were independent of disease activity at the initiation of treatment. Moreover, even in patients with GI tumors, we did not observe an increased occurrence of GI side effects.

We identified one case of digestive perforation in a patient with CRC and quiescent CD who was treated with infliximab six months prior to cancer diagnosis. Perforation occurred upon the first round of bevacizumab as maintenance therapy. However, the frequency of intestinal perforation in our cohort did not seem to be superior to that observed in patients with sporadic CRC [15,16].

While there is evidence of IBD flares under adjuvant hormone therapy [11], reports of the effects of chemotherapy on IBD range from no effects [21] to therapy-associated beneficial outcomes on disease activity [11], which our study confirms.

Among our small cohort, 11 patients with CD-related CRC were treated with bevacizumab, providing data on treatment efficacy that requires careful interpretation. Specifically, in our cohort, patients with IBD-related CRC treated with bevacizumab as first-line therapy showed a modest response to treatment with a median PFS of 5.4 months. Due to the small sample size, survival data are not statistically significant and differ from those of the pivotal trials [22]. Regarding molecular findings, we identified two patients (15%) with *KRAS* mutations. Unfortunately, we could not provide data about MMR status or *BRAF* mutations because this information was not collected at the time of investigation.

We found no studies addressing the activity of bevacizumab-containing regimens in patients with IBD in the literature. However, clinical and biological features of patients with IBD-related CRC that may affect treatment outcomes are well established. CRC in patients with IBD represents less than 2% of all CRC cases and is characterized by poorly differentiated and often synchronous tumors, less frequently involving the rectum. IBD-related CRC accounts for 10% to 15% of deaths in patients with IBD and occurs more frequently in men than sporadic cancers [23]. The molecular characteristics of IBD-associated CRC are suggested to differ between UC and CD patients. In UC, an increased *TP53* mutational load [24] and mutations in the APC/beta-catenin pathway [25] have been described, while microsatellite instability [26] is observed in both UC and CD patients. In contrast to UC, oncogenic pathways in CD remain insufficiently characterized. While some authors suggest similar *BRAF, NRAS, and KRAS* mutational profiles in CD-associated and sporadic CRC [27], the results of a recent meta-analysis investigating both UC and CD-associated CRC described a higher prevalence of *TP53* mutations and a lower prevalence of *KRAS* mutations in IBD-associated CRC than in sporadic CRC [28]. Hence, the molecular characteristics of IBD-associated CRC might affect treatment response and guide future treatment strategies.

The main limitation of the present study lies in its retrospective nature. Indeed, the complex clinical and biological disease variables that may help evaluate the cumulative disease burden of IBD are not captured on a routine basis in the oncology records and, therefore, were not always available for this retrospective analysis. Hence, only surrogate markers, such as time from IBD diagnosis to cancer diagnosis and treatment during the six months before cancer diagnosis were considered. A better characterization of both clinical and biological markers for cumulative IBD activity may help predict cancer risk and potential complications in patients with IBD receiving bevacizumab.

## 5. Conclusions

In conclusion, our study provides real-world evidence that bevacizumab in combination with chemotherapy treatment for cancer patients with quiescent or moderately active IBD is feasible with an acceptable safety profile. Active therapies should not be withheld from patients with quiescent or moderately active IBD, and more clinical data of patients with severe chronic forms of IBD and those that require corticoid-sparing therapies are needed. This study can be used as a basis to implement treatment schemes and strategies in patients suffering from IBD who are going to undergo cancer treatment.

## Figures and Tables

**Figure 1 cancers-14-02914-f001:**
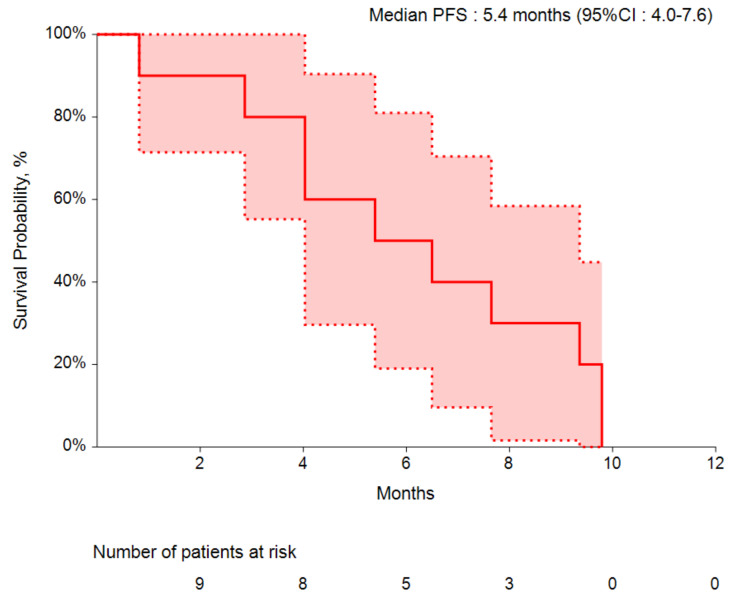
Median progression-free survival (PFS) of patients with IBD-related colorectal cancer treated with chemotherapy and bevacizumab in first line. The area shaded in red represents the 95% confidence interval (CI).

**Table 1 cancers-14-02914-t001:** Baseline characteristics. IBD: inflammatory bowel disease; * IBD medication 6 months before cancer diagnosis consisted of corticosteroids in five patients and infliximab in one patient.

N (%)	All Patients	IBD Remission at Cancer Diagnosis (n = 18)	Moderately Active IBD at Cancer Diagnosis (n = 9)
	(n = 27)		
**Mean age at IBD diagnosis, years (range)**	30.8 (9–59)	28.6 (9–59)	34.8 (12–54)
**Gender**			
** Male**	11 (41%)	7 (39%)	4 (44%)
** Female**	16 (59%)	11 (61%)	5 (56%)
**Type of IBD and involvement**			
** * Ulcerative colitis* **	5 (19%)	5 (28%)	0
** Pancolitis**	2 (40%)	2 (40%)	-
** Indeterminate**	3 (60%)	3 (60%)	-
** * Crohn’s disease* **	22 (81%)	13 (72%)	9 (100%)
** Pancolitis**	5 (23%)	2 (15%)	3 (33%)
** Left-sided colitis**	1 (4%)	1 (8%)	-
** Right-sided colitis**	-	-	-
** Terminal ileum**	3 (14%)	-	3 (33%)
** Ileo-colon**	5 (23%)	2 (15%)	3 (33%)
** Indeterminate**	8 (36%)	8 (62%)	-
**Smoking history**			
** Yes**	5 (19%)	3 (17%)	2 (22%)
** No**	12 (44%)	7 (39%)	5 (56%)
** Unknown**	10 (37%)	8 (44%)	2 (22%)
**Prior surgery for IBD**			
** Yes**	8 (30%)	3 (17%)	5 (56%)
** No**	19(70%)	15 (83%)	4 (44%)
**IBD medication 6 months before cancer diagnosis**	6 * (22%)	1 (6%)	5 (56%)

**Table 2 cancers-14-02914-t002:** Characteristics of cancer and cancer surgery at diagnosis; CUP: carcinoma unknown primary; * type of chemotherapy with bevacizumab in first or second line; ** 5FU: 5-fluorouracil; *** lomustine and everolimus.

	All Patients	Inactive IBD at Cancer Diagnosis	Moderately Active IBD at Cancer Diagnosis
	(n = 27)	(n = 18)	(n = 9)
**Mean age at diagnosis of cancer, years (range)**	44.7 (23–73)	46.5 (25–73)	41.6 (23–54)
**Mean diagnosis time from IBD to cancer, years (range)**	15 (0.5–39)	18 (6–39)	11(0.5–19)
**Performance status**			
** 0**	16 (59%)	9 (50%)	7 (78%)
** 1**	9 (33%)	7 (39%)	2 (22%)
** 2**	1 (4%)	1 (6%)	-
** Unknown**	1 (4%)	1 (6%)	-
**Type of primary cancer**			
**Extraintestinal cancer**			
** Breast**	3 (11%)	1 (6%)	2 (22%)
** Lung**	3 (11%)	3 (17%)	-
** Thyroid**	1 (4%)	1 (6%)	-
** Glioblastoma**	1 (4%)	1 (6%)	-
** CUP**	1 (4%)	-	1 (11%)
**Gastrointestinal cancer**	18 (66%)	12 (67%)	6 (67%)
** Large bowel**	13 (72%)	10 (83%)	3(50%)
** Small bowel**	4 (22%)	1 (8%)	3 (50%)
** Appendix**	1 (6%)	1 (8%)	-
**Disease stage**			
** Metastatic**	16 (59%)	12 (67%)	4 (44%)
** Non-metastatic**	11 (41%)	6 (33%)	5 (56%)
**Radiation included in cancer treatment**			
** Yes**	7 (26%)	6 (33%)	1 (11%)
** No**	19 (70%)	12 (67%)	7 (78%)
** Unknown**	1 (4%)	-	1 (11%)
**Mutated *KRAS* in patients with CRC**	13	10	3
** No**	11(85%)	0	0
** Yes**	2 (15%)	2	0
**Type of surgery for cancer**			
** Right colectomy**	5	4	1
** Left colectomy**	1	1	-
** Total colectomy**	3	2	1
** Proctosigmoidectomy**	5	2	3
** Abdominoperineal resection**	1	-	1
** Others**	10	6	4
** Unknown**	5	4	1
**Type of chemotherapy ***			
** 5FU ** in combination with**	18 (67%)	12 (67%)	6 (67%)
** Oxaliplatin**	8 (44%)	7	1
** Irinotecan**	8 (44%)	3	5
** Oxaliplatin and irinotecan**	1 (6%)	1	0
** 5-FU monotherapy**	1 (56%)	1	0
** Taxanes**	7 (26%)	4 (22%)	3 (33%)
** Others *****	2 (7%)	2 (11%)	0
** Bevacizumab**			
** 1st line**	19 (70%)	13 (72%)	6 (67%)
** 2nd line**	8 (30%)	5 (28%)	3 (33%)
**Mean duration of bevacizumab treatment (months)**	6.7	7.3	5.6
**Reason for treatment discontinuation**			
** Progression**	21 (78%)	14 (78%)	7 (78%)
** Toxicities**	3 (11.1%)	2 (11%)	1 (11%)
** Death**	1 (4%)	1 (6%)	-
** Other**	1 (4%)	-	1 (11%)
** Unknown**	1 (4%)	1 (6%)	-

**Table 3 cancers-14-02914-t003:** Type of cancer depending on underlying IBD type: Crohn’s disease vs. ulcerative colitis; CUP: carcinoma of unknown primary.

	Crohn’s Disease (n = 22)	Ulcerative Colitis (n = 5)
**Mean time between diagnosis of IBD and diagnosis of cancer in years (range)**	13 (0.5–23)	24 (12–39)
**Type of primary cancer**		
** *Extraintestinal cancer* **	**7 (32%)**	**2 (40%)**
Breast	3 (14%)	-
Lung	2 (9%)	1 (20%)
Thyroid	-	1 (20%)
Glioblastoma	1 (5%)	0
CUP	1(5%)	0
** *Gastrointestinal cancer* **	**15 (68%)**	**3 (60%)**
Large bowel	11 (73%)	2 (67%)
Small bowel	4 (27%)	-
Appendix	-	1 (33%)

**Table 4 cancers-14-02914-t004:** Toxicities of bevacizumab and chemotherapy in patients with IBD.

	Whole Population (n = 27)	Inactive IBD at Cancer Diagnosis (n = 18)	Grade	Moderately Active IBD at Cancer Diagnosis (n = 9)	Grade
**Hypertension**	1 (4%)	1 (6%)	2	-	-
**Proteinuria**	2 (7%)	1 (6%)	3	1 (11%)	2
**Epistaxis**	1 (4%)	-	-	1 (11%)	2
**Perforation**	1 (4%)	1 (6%)	5	-	-
**Rectal bleeding**	1 (4%)	1 (6%)	2	-	-
**Bucco-gingival fistula**	1 (4%)	-	-	1 (11%)	2
**Diarrhea**	1 (4%)	-	-	1 (11%)	2
**Neutropenia**	2 (7%)	2 (11%)	3	-	-
**Thrombocytopenia**	1 (4%)	1 (6%)	3	-	-

## Data Availability

All data generated or analyzed during this study are included in this published article.

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
