# Peer review of "Safety and Efficacy of Bevacizumab in Cancer Patients with Inflammatory Bowel Disease"

_cancers, 2022, doi:10.3390/cancers14122914_

Round 1

Reviewer 1 Report

Herrera-Gómez et al present their work entitled "Safety and efficacy of bevacizumab in cancer patients with inflammatory bowel disease".

Although the topic is interesting, some major improvement are warranted:

The introduction needs major revision.

"Regarding 61 medical therapy for IBD, aminosalicylates, corticosteroids, or immunomodulatory agents  such as anti-tumor necrosis factor-alpha (anti-TNF- ) are used": this is not correct. Within the last years, many new medical options were released for IBD patients, also in 2016. This must be adapted and discussed.

"Moreover, a prominent feature of IBD is inflammation-mediated dysregulated neovascularization, whose dysfunctional architecture and increased recruitment of inflammatory cell types contribute to the escalation in inflammatory responses". This is not current understanding of the pathogenesis of IBD and needs to be corrected and explained.

The molecular pathomechanism for e.g. CRC development in IBD is missing and should be explained.

"there is a correlation between CD and hematological and solid non-digestive cancers, which are believed to be a consequence of immunosuppressive therapies." This is also not correct and needs to be adapted. E.g. cancer development is due to chronic inflammation over years, not medication alone. A more differentiated picture of cancer development in IBD needs to be provided.

"This ailment is a chronic inflammatory disorder typically affecting the digestive mucosal tissue of the large bowel in UC and of the large bowel and/or the small intestine". This is not correct. CD is characterized by transmural inflammation while UC is limited to the mucosal layer.

Risk factor for cancer development in IBD is longstanding, active disease during disease course (not only at time of cancer diagnosis). Therefore this information needs to be provided for all IBD patients, including the medication history for all IBD medications applied in the medical history of the patients.

"IBD is a risk factor for the development of gastrointestinal and extra-intestinal cancers." It must be pointed out, that the rist of gastrointestinal cancer is only slightly above a healthy patient and that there are several risk factors for GI cancer. This should be presented in more detail.

All over, the manuscript is not easy to read, e.g. sentences are too long ("Moreover, the concern about the potential side effects of bevacizumab, including intestinal perforation, in patients with IBD and gastrointestinal cancer is based on observations that bevacizumab may affect tumor invasion of gastrointestinal serosa with subsequent necrosis and would thus predispose patients to bowel perforation") and not easy to understand.

Author Response

Review #1: About the introduction needs major revision.

  1. “Regarding medical therapy for IBD, aminosalicylates, corticosteroids, or immunomodulatory agents such as anti-tumor necrosis factor-alpha (anti-TNF- ) are used": this is not correct. Within the last years, many new medical options were released for IBD patients, also in 2016. This must be adapted and discussed

We appreciate the suggestion and we changed for:

Therapies for IBD may consist of untargeted therapies such as aminosalicylates, corticosteroids, immunomodulators and targeted therapies including biologics combined with other general measures such diet programs. Additionally, surgical procedures are proposed if necessary. In non-responders, further, but still experimental therapies are emerging, involving targeted therapies with small molecules, apheresis therapy, interventions that aim to improve the intestinal microbiota, cell therapy and exosome therapy [5].

  1. "Moreover, a prominent feature of IBD is inflammation-mediated dysregulated neovascularization, whose dysfunctional architecture and increased recruitment of inflammatory cell types contribute to the escalation in inflammatory responses". This is not current understanding of the pathogenesis of IBD and needs to be corrected and explained

We appreciate your suggestion. Regarding this part, we believe that we have not explained ourselves adequately, since neovascularization is the consequence of inflammation and is not the pathogenesis of IBD. We corrected our error for:

It is suggested, that the pathogenesis of IBD results from an inappropriate inflammatory response to intestinal microbes in a genetically susceptible host [6]. There are different reports stressing the critical role of neo-angiogenesis in IBD eventually driven by inflammation and the immune response -  a so-called  “immune angiogenesis” [7]. In patients with IBD, a higher density of micro vessels in the intestinal tissue has been demonstrated, as well as higher levels of vascular endothelial growth factor(VEGF) in serum and colonic tissue [8].

  1. The molecular pathomechanism for e.g. CRC development in IBD is missing and should be explained.

We thank the reviewer for pointing out this issue and we agree that it would be useful to delve into the pathogenesis of CRC in patients with IBD. Still, we believe that adding this information could divert the reader from the main objective, which is to assess the safety of bevacizumab in this population group. For this reason, we decided not to add this information, but we emphasize the rest of your suggestions.

  1. "there is a correlation between CD and hematological and solid non-digestive cancers, which are believed to be a consequence of immunosuppressive therapies." This is also not correct and needs to be adapted. g. cancer development is due to chronic inflammation over years, not medication alone. A more differentiated picture of cancer development in IBD needs to be provided.

We appreciate your suggestion. In fact, we have adapted for:

As a consequence of chronic inflammation, patients diagnosed with IBD have a slightly increased risk of developing cancer compared to the general population. Of note, the lifetime risk of developing colorectal cancer (CRC) in patients with IBD in Europe and North America is 1.5 to 2 times higher. Most important risk factors for CRC in patients with IBD are extensive colitis and chronic active disease [9].

 Immunosuppressive therapies and lifestyle factors, such as tobacco consumption, frequently observed in patients with CD, may further contribute to an elevated risk of cancer [9].  Beside CRC, other digestive cancers such as adenocarcinoma of the small intestine, cholangiocarcinoma, intestinal lymphomas, and anal cancers may be related to IBD. Cancers associated with immunosuppression are hematologic cancers, but also skin cancers such as melanoma cancer, basal cell carcinomas, and squamous cell carcinomas; and urinary tract cancers [9]. It goes without saying that independently of possible IBD-related cancers, sporadic cancers remain the most frequent cancers in patients with IBD.

  1. ‘’This ailment is a chronic inflammatory disorder typically affecting the digestive mucosal tissue of the large bowel in UC and of the large bowel and/or the small intestine". This is not correct. CD is characterized by transmural inflammation while UC is limited to the mucosal layer.

We agree and have updated for:

The diagnosis of IBD is typically established in young adults and is based on histological findings. In case of UC, the digestive mucosal tissue of the large bowel is affected. Usually, lesions are arising from the rectum having a continuous extension. In CD, a transmural inflammation is observed that may affect any part of the gastrointestinal tract, but typically develops in the terminal ileum.

  1. Risk factor for cancer development in IBD is longstanding, active disease during disease course (not only at time of cancer diagnosis). Therefore, this information needs to be provided for all IBD patients, including the medication history for all IBD medications applied in the medical history of the patients.

We totally agree with your observation, but previous lines of treatment for IBD were not collected from all patients, given the retrospective nature of the work. We mentioned this limitation in the discussion section. The time between IBD diagnosis and cancer diagnosis was also added to the revised manuscript.

  1. "IBD is a risk factor for the development of gastrointestinal and extra-intestinal cancers." It must be pointed out, that the rist of gastrointestinal cancer is only slightly above a healthy patient and that there are several risk factors for GI cancer. This should be presented in more detail.

We appreciate your suggestion. We have already responded to this suggestion in item No. 4.

In conclusion, we have revised and improved the introduction by updating the indicated treatment methods and we insist on refining the pathogenesis of IBD, as well as providing clearer and more detailed information on the risk factors of IBD, the time until the appearance of the cancer and other features. In addition, the entire manuscript was partially rewritten, and quality checked for the English language. We would like to thank the evaluator again for taking the time to review our manuscript.

Reviewer 2 Report

The authors of the manuscript took up a very interesting research topic. Studies describing the safety of bevacizumab in IBD are low. The activity of this drug, as a humanized IgG1 monoclonal antibody directed against VEGF-A, is of particular interest in this patient group.

The weakness of the study is the small size of the study group.

Taking into account the fact that the potential side effect may be related to the activity, form, duration of the disease. This part of the work needs to be supplemented:

  • Classification of CU disease - left-sided/ right-sided/ pancolitis form.
  • Classification CD - Montreal Classification Disease
  • Activity CD-CDAI,
  • Activity CU - Mayo / T-W classification.
  • In addition, a few sentences in the Introduction require clarification:

  • „Notably, in recent years, the incidence and prevalence of IBD increased in both developed and developing countries”

  • „Consensual definitions of disease severity for IBD are still lacking, and it has been suggested to grade disease by the impact of the disease on the patient, the inflammatory burden and the disease course, thus per- 60 mitting a classification of quiescent, moderately active, and severe disease.”

Author Response

Review #2

We agree with the reviewer that all data including classification of IBD are important to better characterize this pathology, but unfortunately, these data are not captured on a routine basis in Oncology medical records and could not be retrieved for the purpose of the present retrospective analysis. We have explained this limitation in the discussion section since Mayo or Montreal scores could not be derived from the available input.  Also, as noted before, we have revised and modified our introduction (see below). We do hope you find the quality of our work improved

Regarding the major modification in the introduction on:

"In particular, in recent years, the incidence and prevalence of IBD have increased in both developed and developing countries"

We have changed it to:

In Europe and North America the prevalence of IBD ranges from 0.3% to 0.5% in the general population. A stable or decreasing incidence over the past decades is observed. An increase of both prevalence and incidence of IBD has been observed in newly industrialized countries.

We would like to thank the evaluator again for taking the time to review our manuscript.

Reviewer 3 Report

This manuscript is an original article that retrospectively investigated the safety and efficacy of bevacizumab and chemotherapy in metastatic solid cancer patients with inflammatory bowel disease (IBD). The authors showed that neither clinical IBD flares nor IBD-specific complications were observed during bevacizumab treatment, and concluded that bevacizumab combined with chemotherapy was safe in cancer patients with IBD.

This article contains informative information, which will be of interest to researchers and clinicians in the field.

However, the following major and minor issues require clarification:

Major

  1. Most patients with IBD usually maintain some treatment as maintenance therapy even in remission state. It seems strange that the majority of the patients were off treatment during cancer therapy. Please describe medication before and during cancer therapy in detail, including 5ASA and immunomodulators.
  2. Introduction is somewhat redundant and includes information which is little related to the main subject. I recommend that the authors focus on the possible relationship between IBD and bevacizumab as described in the first and second paragraph in the Discussion.

Minor

  1. (Table 2) Please add “Mean” in front of “diagnosis time from IBD to cancer”.
  2. (P10L252) The authors should explain two patients presented with Grade 2 rectal bleeding.

3. Please use an abbreviation of “GI” for “gastrointestinal”.

Author Response

Review #3

We thank the reviewer for pointing this out.

We have reviewed with respect to major changes,

  1. Most patients with IBD usually maintain some treatment as maintenance therapy even in remission state. It seems strange that the majority of the patients were off treatment during cancer therapy. Please describe medication before and during cancer therapy in detail, including 5ASA and immunomodulators.

We revised the patient files and added information about documented systemic IBD-targeted therapies received during and in the 6 months before cancer treatment in the main text:

The documented IBD treatment in the 6 months before cancer diagnosis was systemic corticosteroids in five patients and infliximab in one patient. 5-ASA and topical corticosteroids were additionally used as clinically indicated. Twenty-one patients were off systemic treatment and notably no high-dose corticosteroids (defined as prednisone equivalent of > 20 mg/d) were initiated during cancer therapy.

Unfortunately, due to the retrospective nature of this study and incomplete documentation of the patient files, information about the exact frequency of use of 5-ASA and topical steroids could not be retrieved.

  1. Regarding the second suggestion on introduction, this was corrected with a focus on the relationship of IBD and bevacizumab.

Considering the minor points,

  1. (Table 2) Please add “Mean” in front of “diagnosis time from IBD to cancer”: We agree and have updated.
  2. (P10L252) The authors should explain two patients presented with Grade 2 rectal bleeding: Only one patient presented grade 2 rectal bleeding that was investigated using colonoscopy. After exclusion of disease reactivation, treatment was continued.
  3. Please use an abbreviation of “GI” for “gastrointestinal”: We agree and have updated.

We would like to thank the evaluator again for taking the time to review our manuscript.

Round 2

Reviewer 3 Report

I appreciate that the authors have revised the manuscript according to my suggestions. The manuscript is much improved enough to be accepted.